# Comparison of the epidemiological and clinical fingerprints of Human Granulocytotropic Anaplasmosis and Human Monocytotropic Ehrlichiosis in the United States

Christian Kositz[1,2]*, Larissa Gygax[1,3], Sophie Schudel[1,3], Esther Kuenzli[1,3], Andreas Neumayr[1,3,4]

1 Swiss Tropical and Public Health Institute, Basel, Switzerland, 2 Clinical Research Department, Faculty of Infectious and Tropical Diseases, London School of Hygiene & Tropical Medicine, London, United Kingdom, 3 University of Basel, Basel, Switzerland, 4 Department of Public Health and Tropical Medicine, College of Public Health, Medical and Veterinary Sciences, James Cook University, Queensland, Australia

* christian.kositz@swisstph.ch

## Abstract

### Background

Human granulocytotropic anaplasmosis (HGA), caused by *Anaplasma phagocytophilum,* and human monocytotropic ehrlichiosis (HME), caused by *Ehrlichia chaffeensis*, are tick-borne zoonoses. The vast majority of *Anaplasma phagocytophilum* infections reported worldwide are from the United States. *Ehrlichia chaffeensis* infections are reported exclusively from North America, with the majority reported from the United States. We have recently summarized the available data on HGA and HME in the form of two systematic reviews. We compared data from these two systematic reviews to objectify differences in epidemiological and clinical presentation between the two diseases in the United States.

### Methods

From our recently published systematic reviews on HGA and HME, «best evidence data» were extracted and comparatively analyzed. Cases were included if (i) they had a high level of diagnostic certainty (i.e., diagnosed by PCR, culture, immunostaining of tissue, or paired IgG IFA serology), (ii) individual clinical data were available, (iii) no concomitantly present coinfection(s) were reported, and (iv) the infection was acquired in the United States.

### Results

HME cases were statistically more frequent immunocompromised and younger and had gastrointestinal symptoms, hepatosplenomegaly and elevated liver function tests (LFT) levels. HGA cases were statistically more frequent presenting with pulmonary

**Data availability statement:** The data underlying the results presented in the study are available from the supplementary files from these publications Schudel S, Gygax L, Kositz C, Kuenzli E, Neumayr A. Human granulocytotropic anaplasmosis—A systematic review and analysis of the literature. PLoS Negl Trop Dis. 2024 Aug 5;18(8):e0012313. and Gygax L, Schudel S, Kositz C, Kuenzli E, Neumayr A. Human monocytotropic ehrlichiosis—A systematic review and analysis of the literature. PLoS Negl Trop Dis. 2024 Aug 2;18(8):e0012377.

**Funding:** The author(s) received no specific funding for this work.

**Competing interests:** The authors have declared that no competing interests exist.

symptoms and encephalitis. Hospitalization and a fatal outcome was more frequent in HME. No differences in fatal outcome were observed between immunocompetent and immunocompromised HME cases.

## Conclusion

Although HGA and HME show similarities, there are statistically significant differences in terms of their clinical patterns and outcomes and a not yet described difference in the affected age pattern.

## Introduction

Human granulocytotropic anaplasmosis (HGA), caused by *Anaplasma phagocytophilum* and human monocytotropic ehrlichiosis (HME), caused by *Ehrlichia chaffeensis*, are tick-borne zoonoses. *Anaplasma phagocytophilum* infections are reported from all continents except Antarctica, with the vast majority of cases being reported from North America, particularly the United States [1,2]. *Ehrlichia chaffeensis* infections are reported exclusively from North America, again with the vast majority of cases reported from the United States. HGA is transmitted by the blacklegged tick (*Ixodes scapularis*) and the western blacklegged tick (*Ixodes pacificus*). HME is primarily transmitted by the Lone Star tick (*Amblyomma americanum*). The primary host animal reservoir hosts sustaining the enzootic cycles of *A. phagocytophilum* and *E. chaffeensis* are the white-footed mouse (*Peromyscus leucopus*) and the white-tailed deer (*Odocoileus virginianus*), respectively [3,4].

Since HGA and HME have been recognized as emerging infections with over the years steadily increasing incidences in the United States [5–7], both diseases are now notifiable in all states (Fig 1).

For example 2019, 5655 HGA cases and 2093 HME cases were reported [8,9]. In 2020, the number of reported HGA and HME cases decreased to 3637 and 1178 respectively due to the COVID-19 pandemic [8,9]. The latest available data from 2022 shows an increase compared to the pre-pandemic situation with 5651 reported HGA cases, while the number of HME cases remains below the pre-pandemic level with 1570 reported cases [8,9].

HGA and HME present with similar, unspecific, flu-like symptoms, including fever, muscle aches, gastrointestinal symptoms and headache and mostly show a benign and self-limiting course, even if left untreated. However, both infections can lead to serious and even life-threatening complications. The presumptive diagnosis is mostly based on potential tick exposure/ history of a tick bite combined with a compatible clinical picture, cytopenia (primarily thrombocytopenia and leukopenia) and elevated liver function tests. Antimicrobial treatment is started upon suspicion, as delayed treatment is associated with an increased risk of progression to severe illness [10]. The treatment of choice for HGA and HME is doxycycline or tetracycline and the clinical response is generally rapid [2,11].

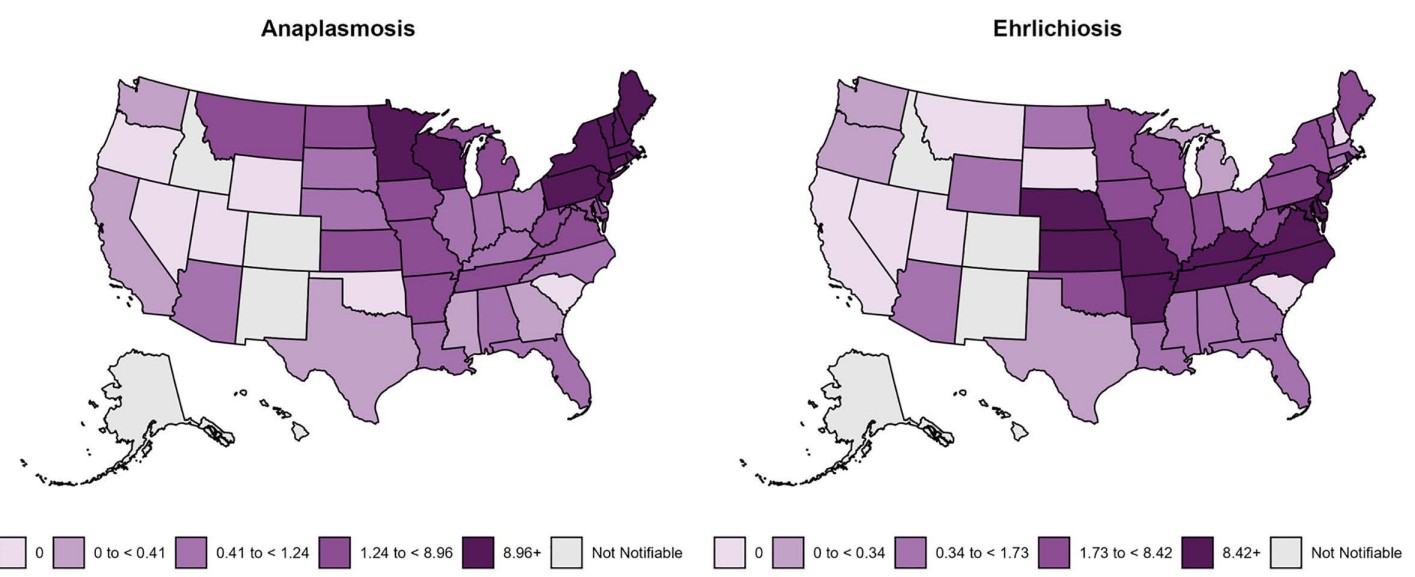

**Fig 1. Annual incidence (per million population) of reported anaplasmosis and ehrlichiosis in the United States for 2022 (adapted from [8,9], materials developed by CDC).**

United States surveillance data show hospitalization and overall case fatality rates of 31% and 0.3% for HGA [12] and 57% and 1% for HME [6,13].

As most of the available data on HGA and HME come from case reports, case series and retrospective studies, while prospective studies and clinical trials are lacking, some aspects of the two diseases remain incompletely understood. To obtain a clearer picture and to better characterize the two diseases, we recently systematically reviewed the available literature and published our findings in two separate systematic reviews [2,11]. In this paper, we use the best-evidenced data from each of these two systematic reviews to comparatively analyze and describe the epidemiological and clinical fingerprint of the two diseases in the United States.

## Methods

From each of the compiled datasets of our systematic reviews on HGA [2] and HME [11] we extracted a «best-evidence data subset» for each disease. These reviews included all cases of anaplasmosis (published up to 27/Apr/2023) and ehrlichiosis (published up to 26/June/2023) published in English, German, French, Italian or Spanish. Cases were eligible to be included if (i) they had a high level of diagnostic certainty, (ii) individual clinical data were available, (iii) no concomitantly present coinfection(s) were reported, and (iv) the infection was acquired in the United States (see S1 Fig). A high level of diagnostic certainty was defined as diagnosed by (i) PCR, culture and/or immunostaining of biopsy/autopsy/ tissue or (ii) paired IgG IFA serology (= all cases labelled as A+ or A in our systematic reviews [2,11]). We limited our analysis to the United States because only this region has sufficient high-quality data and HME is not endemic outside North America and plotted the data geographically (see S2 Fig).

Data was categorized into demographic, clinical outcome, laboratory and symptoms, in addition we split the dataset into immunocompromised and immunocompetent cases. Categorical variables were compared using Chi-squared test for n ≥ 5 and Fisher's exact test for n < 5, respectively. For comparing continuous variables, the student's t-test was used for parametric data; for non-parametric data, the Kruskal-Wallis test was used. The statistical analysis was done using R Version 2023.06.0 Build 421. To evaluate whether the two diseases have a different age

distribution through life, a two-sample Kolmogorov-Smirnov test was used after testing for normality using a Shapiro test. Fig 1 was adapted from CDC, and Fig 2 which shows the epidemiological pattern of infections according to age, age and gender, and age, gender and immunosuppression were both created using ggplot2 in R Version 2023.06.0 Build 421.

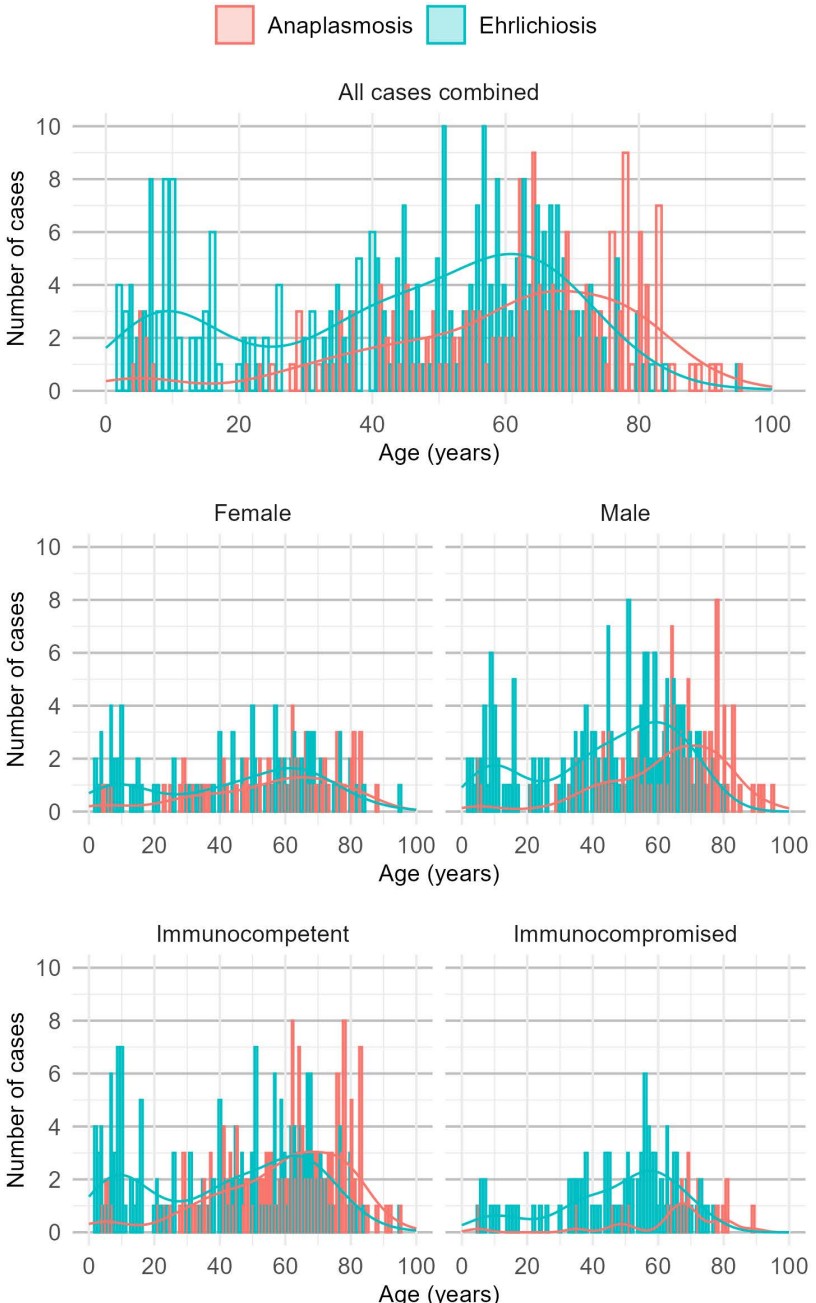

**Fig 2. Epidemiological distribution of the analyzed 176 HGA and 272 HME cases according to age, sex, and immune status using US data.**

## Results

Of the 3019 HGA and the 1260 HME cases compiled in our systematic reviews on HGA [2] and HME [11], 176 HGA and 272 HME cases met the inclusion criteria and were included in the analysis. Table 1 lists the demographic details of these cases, which showed a significant difference in the median ages affected between HGA and HME in immuno-competent (62 vs 47 years of age, p < 0.001), and immunosuppressed (68 vs 52 years of age, p = 0.01). Table 2 shows a comparison of symptoms and signs, with rash being more common in immunocompetent HME cases than HGA cases (26% vs 9.7%, p < 0.001) and no difference between immunocompromised. Furthermore, gastrointestinal symptoms were significantly more common in immunocompetent HME cases than HGA cases (58% vs 32, p < 0.001), specifically hepatosplenomegaly (16% vs 4%, p = 0.002) whereas pulmonary symptoms were more common in HGA than in HME in both immunocompetent (27% vs 16%, p = 0.024) and immunosuppressed (50% vs 13%, p < 0.001). The difference in liver pathology is also reflected in the laboratory markers such as AST, ALT being significantly more often elevated (35% vs 58%, p < 0.001, and 28% vs 46%, p < 0.001, respectively) in immunocompetent HME cases than HGA cases shown in Table 3, which compares all available laboratory parameters. Table 4 compares clinical parameters such as outcome, hospitalization and complications and shows a significantly higher proportion of immu-nocompetent HME cases being hospitalized than HGA cases (62% vs 81%, p < 0.001) with no such difference seen in immunocompromised (p > 0.99). The overall number of complications was higher in HME (64% vs 46%, p = 0.001) if the person was immunocompetent and reversed with more complications in HGA in immunocompromised (83% vs 39%, p < 0.001). Fatal outcomes (16% vs 4.2%, p < 0.001) and sHLH (13% vs 1.9%, p < 0.001) were signifantly more often in HME than HGA, though no such difference was seen in immunocompromised with a p = 0.16 in sHLH, and a p > 0.99 in fatal outcomes, respectively. Fig 2 shows the graphical distribution of these cases according to age, sex, and immune status. The Kolmogorov-Smirnov test found evidence for a difference in age distribution between ehrlichi-osis and anaplasmosis with a p-value of <0.001

**Table 1. Comparison of the demographic data of the analyzed 176 HGA and 272 HME cases.**

| | Human Granulocytotropic Anaplasmo-sis (HGA) N = 176 | | Human Monocytotropic Ehrlichiosis (HME) N = 272 | | Comparison HGA vs HME p-value | |
| --- | --- | --- | --- | --- | --- | --- |
| | Immuno-competent | Immuno-compromised | Immuno-competent | Immuno-compromised | Immuno-competent | Immuno-compromised |
| **No. of cases** [n/N (%)] | 153/176 (86) | 23/176 (14) | 175/272 (64) | 97/272 (36) | 0.05 | <0.001† |
| **Age group** [n/N (%)] | | | | | | |
| <20 | 7/147 (4.8) | 1/22 (4.5) | 50/175 (29) | 12/96 (13) | | |
| 20–40 | 17/147 (12) | 1/22 (4.5) | 18/175 (10) | 17/96 (18) | | |
| 40–60 | 38/147 (26) | 3/22 (14) | 50/175 (29) | 41/96 (43) | | |
| >60 | 85/147 (58) | 17/22 (77) | 57/175 (33) | 26/96 (27) | | |
| **Age** [median (IQR)] | 62 (45, 76) | 68 (63, 72) | 47 (16, 65) | 52 (37, 61) | <0.001 | 0.01†† |
| N unknown | 6 | 1 | 0 | 1 | | |
| **Sex** [n/N (%)] | | | | | | |
| Female | 60/147 (41%) | 9/23 (39) | 76/175 (43) | 21/97 (22) | 0.72 | 0.11† |
| Male | 87/147 (59%) | 14/23 (61) | 99/175 (57) | 76/97 (78) | | |

† Pearson's Chi-squared test.

†† Kruskal-Wallis rank sum test

**Table 2. Comparison of the symptoms and signs of the analyzed 176 HGA and 272 HME cases.**

| | Human Granulocytotropic Anaplasmosis (HGA) N = 176 | | Human Monocytotropic Ehrlichiosis (HME) N = 272 | | Comparison HGA vs HME p-value[†] | |
|---|---|---|---|---|---|---|
| | Immuno-competent N = 153 n/N (%) | Immuno-compromised N = 23 n/N (%) | Immuno-competent N = 175 n/N (%) | Immuno-compromised N = 97 n/N (%) | Immuno-competent | Immuno-compromised |
| **Fever** | 120/131 (92) | 20/22 (91) | 164/175 (94) | 95/97 (98) | 0.63 | 0.16 |
| **Malaise/ Fatigue** | 55/124 (44) | 11/22 (50) | 48/175 (27) | 24/97 (25) | 0.004 | 0.037 |
| **Rash** | 12/124 (9.7) | 1/22 (4.5) | 46/175 (26) | 11/97 (11) | <0.001 | 0.46[††] |
| **Headache** | 42/124 (34) | 7/22 (32) | 75/175 (43) | 46/97 (47) | 0.15 | 0.27 |
| **Myalgia** | 55/124 (44) | 7/22 (32) | 63/175 (36) | 20/97 (21) | 0.18 | 0.4 |
| **Arthralgia** | 12/124 (9.7) | 1/22 (4.5) | 13/175 (7.4) | 7/97 (7.2) | 0.63 | >0.99[††] |
| **Gastrointestinal symptoms** | 40/124 (32) | 7/22 (32) | 101/175 (58) | 46/97 (47) | <0.001 | 0.27 |
| Nausea | 20/124 (16) | 3/22 (14) | 41/175 (23) | 27/97 (28) | 0.16 | 0.28[††] |
| Vomiting | 13/124 (10) | 3/22 (14) | 36/175 (21) | 20/97 (21) | 0.031 | 0.56[††] |
| Abdominal pain | 12/124 (9.7) | 2/22 (9.1) | 37/175 (21) | 6/97 (6.2) | 0.013 | 0.6[††] |
| Diarrhea | 15/124 (12) | 4/22 (18) | 25/175 (14) | 18/97 (19) | 0.73 | >0.99[††] |
| Anorexia | 12/124 (9.7) | 1/22 (4.5) | 28/175 (16) | 5/97 (5.2) | 0.16 | >0.99[††] |
| **Hepatosplenomegaly** | 5/124 (4.0) | 0/22 (0) | 28/175 (16) | 4/97 (4.1) | 0.002 | >0.99[††] |
| **Pulmonary symptoms** | 34/124 (27) | 11/22 (50) | 28/175 (16) | 13/97 (13) | 0.024 | <0.001 |
| Dyspnea | 28/124 (23) | 10/22 (45) | 9/175 (5.1) | 5/97 (5.2) | <0.001 | <0.001 |
| Cough | 12/124 (9.7) | 7/22 (32) | 21/175 (12) | 11/97 (11) | 0.66 | 0.037 |
| **Impaired consciousness** | 28/124 (23) | 5/22 (23) | 48/175 (27) | 17/97 (18) | 0.42 | 0.79 |
| **Meningeal symptoms** | 8/124 (6.5) | 0/22 (0) | 9/175 (5.1) | 1/97 (1.0) | 0.82 | >0.99[††] |
| **Cardiovascular symptoms** | 23/153 (15) | 6/23 (26) | 33/175 (19) | 15/97 (15) | 0.44 | 0.37 |

[†] Pearson's Chi-squared test; [††] Fisher's exact test; Differences in denominators are caused by the lack of standardized reporting;

## Discussion

In this study we provide a comparative analysis of human granulocytotropic anaplasmosis (HGA) and human monocytotropic ehrlichiosis (HME) in the United States, focusing on the clinical presentations, laboratory findings, and outcomes of the two diseases. The aim of this comparative analysis was to identify potentially helpful patterns for the clinical differentiation of the two infections.

The reported opportunistic nature of *Ehrlichia* in humans [11,14,15] is reflected in the considerably higher rate of immuno-compromised patients among HME cases (36%) compared to HGA cases (14%) (Table 1). For this reason, all comparative analyses of the two diseases were always carried out separately for the immunocompetent and immunosuppressed cases.

An interesting finding of our study, which has not been reported previously, is the difference in age distribution between the two infections. HME appears to affect a younger cohort, with a median age of 47 years compared to 62 years in immunocompetent HGA. This effect is very likely caused by the additional peak observed in children and adolescences in HME compared to HGA (Fig 2). The observed biphasic age distribution in HME is independent of gender and, as immunosuppression primarily affects older patients, is plausibly less pronounced in the immunocompromised group. However, we cannot answer whether this finding reflects actual differences in, e.g., tick exposure patterns or biological susceptibility to *Ehrlichia* and *Anaplasma* infections in the different age groups or whether the effect is due selection bias in this retrospective analysis of secondary data.

In terms of symptoms, immunocompetent HME cases were statistically significantly more likely to present with skin rash and gastrointestinal symptoms, particularly abdominal pain and hepatosplenomegaly, than HGA cases. This is

**Table 3.** Comparison of the laboratory parameters of the analyzed 176 HGA and 272 HME cases.

| | Human Granulocytotropic Anaplasmosis (HGA) N=176 | | Human Monocytotropic Ehrlichiosis (HME) N=272 | | Comparison HGA vs HME | |
|---|---|---|---|---|---|---|
| | Immuno-competent N=153 | Immuno-compromised N=23 | Immuno-competent N=175 | Immuno-compromised N=97 | Immuno-competent | Immuno-compromised |
| | n/N (%) | n/N (%) | n/N (%) | n/N (%) | p-value[†] | p-value[†] |
| **Anemia** | 41/153 (27) | 4/23 (17) | 66/175 (38) | 31/97 (32) | 0.047 | 0.21[††] |
| **Leukopenia** | 73/153 (48) | 4/23 (17) | 97/175 (55) | 36/97 (37) | 0.2 | 0.087[††] |
| **Thrombocytopenia** | 102/153 (67) | 9/23 (39) | 132/175 (75) | 50/97 (52) | 0.1 | 0.4 |
| **Elevated LFTs** | 75/153 (49) | 13/23 (57) | 119/175 (68) | 31/97 (32) | <0.001 | 0.05 |
| **AST elevated** | 53/153 (35) | 9/23 (39) | 102/175 (58) | 25/97 (26) | <0.001 | 0.31 |
| **ALT elevated** | 43/153 (28) | 6/23 (26) | 81/175 (46) | 21/97 (22) | <0.001 | 0.86 |
| **CRP or ESR elevated** | 24/153 (16) | 6/23 (26) | 12/175 (6.9) | 2/97 (2.1) | 0.018 | <0.001[††] |
| **CK elevated** | 8/153 (5.2) | 1/23 (4.3) | 10/175 (5.7) | 0/97 (0) | >0.99 | 0.19[††] |
| | median (IQR) | median (IQR) | median (IQR) | median (IQR) | p-value[#] | p-value[#] |
| **Hemoglobin Hb (g/dl)** | 10.5 (8.8–12.9) | 10.2 (9.0–11.5) | 10.6 (7.9–11.7) | 9.1 (7.8–11.4) | 0.35 | 0.65 |
| N unreported | 106 | 21 | 106 | 69 | | |
| **Leukocytes (10³/µl)** | 3.5 (2.4–5.7) | 4.1 (2.7–14.8) | 3.1 (1.8–5.1) | 2.5 (1.6–2.9) | 0.12 | 0.1 |
| N unreported | 71 | 17 | 52 | 63 | | |
| **Thrombocytes (10³/µl)** | 60 (26–105) | 59 (26–82) | 56 (35–90) | 40 (25–85) | 0.93 | 0.86 |
| N unreported | 66 | 16 | 47 | 54 | | |
| **AST (U/l)** | 89 (53–167) | 171 (80–173) | 176 (96–566) | 358 (139–650) | <0.001 | 0.065 |
| N unreported | 97 | 14 | 69 | 70 | | |
| **ALT (U/l)** | 73 (40–120) | 65 (44–147) | 151 (76–268) | 123 (73–284) | <0.001 | 0.11 |
| N unreported | 103 | 17 | 89 | 73 | | |

[†] Pearson's Chi-squared test; [††] Fisher's exact test, [#] Kruskal-Wallis rank sum test;. AST, aspartate transferase; ALT, alanine transferase; CRP, c-reactive protein; CK, creatine kinase; ESR, erythrocyte sedimentation rate; LFTs, liver function tests.

also reflected in statistically significantly higher transaminase levels in HME compared to HGA. HGA cases, irrespective of immune status, on the other hand showed a statistically significant association with malaise/fatigue and pulmonary symptoms, specifically dyspnoea, as previously described by Ismail et al. [3]. The lack of a significant difference of rash between immunocompromised HGA and HME cases suggests that the host's immune response contributes to this symptom (Table 2).

Leukopenia, anaemia, and thrombocytopenia are reported to be common in both infections [3,15]. This general cytopenia is also found in our data set. Although anaemia was statistically more common in HME than in HGA, we found no differences in the median values of haemoglobin, leukocytes, and thrombocytes when comparing the two diseases (Table 3). Cytopenia(s) therefore do not appear to be helpful for distinguishing between the two diseases, while the observation of elevated liver function tests would raise suspicion of HME rather than HGA. Although statistically significant, the significance of the observations of a higher frequency of elevated CRP or ESR in HGA compared to HME is limited by the very small data set, which prevented a meaningful quantitative analysis.

The statistically significant higher rates of hospitalization, complications and fatal outcome of HME compared to HGA presents HME as the overall more dangerous infection for immunocompetent patients (Table 4). Interestingly, when looked at separately, the two diseases and their respective immune status, immunocompromisation had no effect on hospitalization, and fatal outcome rate (Table 4). Although immunodeficiency is a predisposition to HME, it does not appear to have an impact on the clinical outcome.

**Table 4. Comparison of hospitalization rate, occurrence of complication(s) and outcome of the analyzed 176 HGA and 272 HME cases.**

| | Human Granulocytotropic Anaplasmosis (HGA) N = 176 | | Human Monocytotropic Ehrlichiosis (HME) N = 272 | | Comparison HGA vs HME | |
|---|---|---|---|---|---|---|
| | Immuno-competent N = 153 | Immuno-compromised N = 23 | Immuno-competent N = 175 | Immuno-compromised N = 97 | Immuno-competent | Immuno-compromised |
| | n/N (%) | n/N (%) | n/N (%) | n/N (%) | p-value[†] | p-value[†] |
| **Hospitalization** | 95/153 (62) | 22/23 (96) | 142/175 (81) | 94/97 (97) | <0.001 | >0.99[††] |
| **Occurrence of complication(s)** | 70/153 (46) | 19/23 (83) | 112/175 (64) | 38/97 (39) | 0.001 | <0.001 |
| Multiorgan failure | 15/153 (9.8) | 6/23 (26) | 24/175 (14) | 11/97 (11) | 0.36 | 0.14 |
| Acute kidney failure | 21/153 (14) | 2/23 (8.7) | 21/175 (12) | 15/97 (15) | 0.76 | 0.73[††] |
| Circulatory failure/shock | 9/153 (5.9) | 4/23 (17) | 9/175 (5.1) | 6/97 (6.2) | 0.96 | 0.21[††] |
| ARDS | 15/153 (9.8) | 3/23 (13) | 27/175 (15) | 16/97 (16) | 0.18 | >0.99[††] |
| sHLH | 6/153 (4.2) | 0/23 (0) | 28/175 (16) | 12/97 (12) | <0.001 | 0.16[††] |
| **Fatal outcome** | 3/153 (1.9) | 4/23 (17) | 22/175 (13) | 12/97 (12) | <0.001 | >0.99[††] |

[†] Pearson's Chi-squared test; [††] Fisher's exact test

ARDS, acute respiratory distress syndrome; sHLH, secondary hemophagocytic lymphohistiocytosis

Among the analyzed complications secondary hemophagocytic lymphohistiocytosis (sHLH) was significantly more often associated with HME (Table 4) than with HGA [16]. As noted in our review of HME [11], we have not found any other pathogen in the literature that has a similarly high association rate with HLH as *Ehrlichia*. Thus, sHLH appears to be a signature complication of HME and therefore of differential diagnostic value, but does occur in HGA. No statistically significant differences between HGA and HME were observed regarding the occurrence of multiorgan failure, acute kidney failure, circulatory failure/shock or ARDS (Table 4). Regarding the latter, the statistically significant association of pulmonary symptoms in HGA (Table 2) does not appear to be associated with complicating ARDS. Unfortunately, the data from these case reports were insufficient to provide a reliable estimate on the difference of cases with possible encephalitis between the two diseases, and therefore this point remains unresolved.

The limitations of our study are the retrospective design and the overall limited number of cases and data and its lack of standardized collection. No retrospective analysis of data can eliminate the possibility that the data analyzed contains some reporting/publication bias that distorts the picture and the differential missingness of reported symptoms across the two diseases could bias the estimated proportions. It therefore remains to be seen whether prospective studies, should such studies ever be carried out, will confirm or contradict the correlations found. Furthermore, the data is limited to the USA, as hardly any ehrlichiosis cases have been reported outside this geographical distribution.

We conclude that in the published literature HGA and HME differ significantly in terms of the age groups affected. Both diseases show very similar and largely overlapping clinical pictures and can mimic each other. Nevertheless, the differences between HGA and HME described above may be helpful for clinicians to distinguish between the two diseases. Future research may help to elucidate the pathophysiological mechanisms of these observed clinical differences.

## Supporting information

**S1 Fig.** Allocation and selection of cases from the Schudel et al, and Gygax et al with the addition of selection for cases for the USA for analysis.
(TIF)

**S2 Fig.** Geographic distribution of the analyzed 176 HGA and 272 HME cases.
(TIF)

## Author contributions

**Conceptualization:** Christian Kositz, Larissa Gygax, Sophie Schudel, Esther Kuenzli, Andreas Neumayr.

**Data curation:** Christian Kositz, Larissa Gygax, Sophie Schudel.

**Formal analysis:** Christian Kositz, Andreas Neumayr.

**Methodology:** Christian Kositz, Larissa Gygax, Sophie Schudel, Esther Kuenzli, Andreas Neumayr.

**Supervision:** Esther Kuenzli, Andreas Neumayr.

**Visualization:** Christian Kositz.

**Writing – original draft:** Christian Kositz.

**Writing – review & editing:** Christian Kositz, Larissa Gygax, Sophie Schudel, Esther Kuenzli, Andreas Neumayr.

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
