## [Decision Letter · Decision Letter 0]

7 May 2025

Dear Dr. Kositz,

Thank you for submitting your manuscript to PLOS ONE. After careful consideration, we feel that it has merit but does not fully meet PLOS ONE’s publication criteria as it currently stands. Therefore, we invite you to submit a revised version of the manuscript that addresses the points raised during the review process.

We look forward to receiving your revised manuscript.

Kind regards,

Yong Qi

Academic Editor

PLOS ONE

Journal Requirements:

Reviewers' comments:

Reviewer's Responses to Questions

**Comments to the Author**

1. Is the manuscript technically sound, and do the data support the conclusions?

Reviewer #1: Partly

Reviewer #2: Yes

2. Has the statistical analysis been performed appropriately and rigorously?

Reviewer #1: Yes

Reviewer #2: I Don't Know

3. Have the authors made all data underlying the findings in their manuscript fully available?

Reviewer #1: Yes

Reviewer #2: Yes

4. Is the manuscript presented in an intelligible fashion and written in standard English?

Reviewer #1: No

Reviewer #2: Yes

Reviewer #1: Kositz et al. compared reported clinical and epidemiological data for anaplasmosis and ehrlichiosis cases in the United States with the aim of identifying differences in clinical presentation of these two tick-borne illnesses. This summarization provides characterization of clinical and epidemiological features. Further details and rationale are warranted to improve this study. The inclusion criteria for cases in the systematic reviews that forms the basis of these comparisons is not given. Thus, it is unclear how generalizable these features are to anaplasmosis and ehrlichiosis cases more broadly. Also, given that effective treatment is the same for both diseases, further explanation should be given on the clinical utility in differentiating these cases; Lines 164-165 indicate this differentiation is one of the aims of the comparison. In terms of the statistical analysis broadly, why did the authors choose to compare proportions instead of calculating odds ratios. Odds ratios are the standard analyses performed for this type of data and they also produce confidence intervals which are useful for quantifying uncertainty.

Below are some additional points that would benefit from further explanation or synthesis.

Lines 57-59. The conclusion section of the Abstract could be reworded to provide a synthesis of results instead of restating results. I would also caution authors in highlighting statistically significant results without assessing biological or clinical significance of such results throughout the manuscript.

Line 82. Please indicate why 2019-2022 were the only years discussed in terms of case counts? Both diseases were nationally notifiable since 1999.

Lines 95-96. Given that effective treatment for both illness is the same, how important is differentiating anaplasmosis and ehrlichiosis cases clinically? Improving this rationale would help frame the analyses and usefulness of the results.

Lines 110-111. Please provide more details on the systematic reviews as these form the basis for analyses here. For example, how were cases included in these reviews, what was the spatial and temporal scale of included case data, what information was included in each review?

Line 113. Coinfections are not often tested for in clinical settings and not necessarily reportable. However, coinfections potentially impact length and clinical course of the infections. How could the unintentional inclusion of coinfected cases have on the analysis?

Line 123. How were parametric and non-parametric data identified?

Lines 124-127. The results of this Kolmogorov-Smirnov test for differences in age structure does not appear to have been reported. Table 1 presents the results from a Pearson Chi-square test of differences in median age which is not the correct test for this comparison.

Line 128. This statement appears to refer to data presented in Figure 2. However, case data stratified by age, gender, and immunosuppression status is not presented. Please reword this statement or update Figure 2 to provide concordance between methods and presented results.

Lines 132-138. Please expand the Results beyond listing the tables and figure included. Highlight main takeaways presented. Some of this is currently in the Discussion.

Lines 132-133. A relatively large proportion of cases reported in the systematic reviews were not included in this analysis. Which inclusion criteria of the study removed most of these cases? Including a flow chart could illustrate the inclusion/exclusion process for the study.

Lines 166-170. It does not appear that comparing clinical and epidemiolocal data by immune status between anaplasmosis and ehrlichiosis cases was defined as a goal a prior to analyses. As such, the significance testing stratified by immune status (row 1 results presented in Table 1) would represent an ad hoc test, not the primary test. The significance of a Pearson’s chi-squared test on the 2x2 contingency table of immune status vs. case type should be presented. The results of these ad-hoc tests could still be presented, but with a correction for multiple comparisons.

Figure 1. I would suggest presenting the data as bar charts instead of the current density plots. Then it is easier for readers to compare case counts presented in Table 1 with those presented in the figure.

Tables 2-4. Given that the presence and number of asterisks are often used to denote statistical significance, I would suggest using different symbols to denote Fisher’s exact and Kruskal-Wallis tests in Tables.

Table 2. Please explain why the denominators vary across rows. For example, of the 153 immunocompetent cases of HGA, why is the denominator 124 for most symptoms? Was it because this data was not available?

Please review wording throughout the manuscript as there are several grammatical errors. While they do not impair the understanding of the text, they reduce the appearance of high-quality work.

Reviewer #2: This study addresses an important clinical topic; however, the review of the existing literature is insufficient. Several recent and relevant studies have not been cited, which limits the comprehensiveness and accuracy of the background presented. A more thorough literature review is necessary to appropriately contextualize the study findings and conclusions.

1) The term "Fingertips" in the title is misleading and should be modified accordingly. It is unclear what the authors intend to convey—perhaps they mean "characteristics"?

2) In the Introduction, line 66 is missing a reference. Additionally, the authors have not adequately reviewed the existing literature. A recent systematic review on this topic should be referenced and acknowledged: https://pubmed.ncbi.nlm.nih.gov/35889152/

3) Line 76: Both diseases are notifiable in all 50 states in the USA—not “in the most.”

4) Lines 82–87: This section is not particularly illustrative. Moreover, epidemiological data are continuously changing. I suggest removing this part.

5) Line 88: The above-mentioned systematic review also identified gastrointestinal symptoms as common in HGA, and this important detail should be added.

6) Line 99: Please include a more recent reference such as: https://academic.oup.com/cid/advance-article/doi/10.1093/cid/ciaf171/8104456

7) Table 4 reports 29 cases of encephalitis, which appears to be an unusually high number. Please provide a reference for these cases. Encephalitis in HGA is considered extremely rare. Additional clarification or justification is needed—see the following reference: https://pubmed.ncbi.nlm.nih.gov/37489389/

8) Discussion, lines 212–218: Please reference the recent systematic review on tick-borne infection-associated hemophagocytic lymphohistiocytosis (HLH), which provides critical insights into this life-threatening complication: https://pubmed.ncbi.nlm.nih.gov/38525759/. Anaplasma can cause secondary HLH, as described in this review, and the statement in this section should be revised to reflect this evidence

**Do you want your identity to be public for this peer review?** For information about this choice, including consent withdrawal, please see our Privacy Policy

Reviewer #1: No

Reviewer #2: No

---

## [Author Response · Author response to Decision Letter 1]

21 Jun 2025

We cordially want to thank the two reviewers for taking their time to review our manuscript, their critical eyes, and their constructive remarks!

In the following, we will address the points listed by the two reviewers in chronological order and also address what we adapted in the manuscript following the comments from both reviewers.

Reviewer #1:

Kositz et al. compared reported clinical and epidemiological data for anaplasmosis and ehrlichiosis cases in the United States with the aim of identifying differences in clinical presentation of these two tick-borne illnesses. This summarization provides characterization of clinical and epidemiological features. Further details and rationale are warranted to improve this study. The inclusion criteria for cases in the systematic reviews that forms the basis of these comparisons is not given. Thus, it is unclear how generalizable these features are to anaplasmosis and ehrlichiosis cases more broadly. Also, given that effective treatment is the same for both diseases, further explanation should be given on the clinical utility in differentiating these cases;

1. Lines 164-165 indicate this differentiation is one of the aims of the comparison. In terms of the statistical analysis broadly, why did the authors choose to compare proportions instead of calculating odds ratios. Odds ratios are the standard analyses performed for this type of data and they also produce confidence intervals which are useful for quantifying uncertainty.

RESPONSE: We do recognize that it is an option to use odds ratios, however we decided to compare proportions as this might be more helpful to clinicians to know what is significantly more common in one disease, as many of the symptoms and signs overlap or show no difference, especially helpful in areas where the necessary laboratory testing is not available.

Below are some additional points that would benefit from further explanation or synthesis.

2. Lines 57-59. The conclusion section of the Abstract could be reworded to provide a synthesis of results instead of restating results. I would also caution authors in highlighting statistically significant results without assessing biological or clinical significance of such results throughout the manuscript.

RESPONSE: We are unfortunately limited by the word count for the abstract, however rephrased it, but does not lose its importance for clinicians. However, is not quite clear to us, which part of our document does not caution the highlighting of significant results. Specifically, we are pointing the limitations of the data set, which being retrospective observational data from primarily case reports, as well as discrepancies between different presentation, and the need for future research to better understand the minutiae of the pathophysiological differences.

3. Line 82. Please indicate why 2019-2022 were the only years discussed in terms of case counts? Both diseases were nationally notifiable since 1999.

RESPONSE: The years were chosen, as both diseases show an increase of cases over time and the time frame nicely present the impact of the COVID 19 pandemic, with HGA achieving pre-pandemic numbers and HME lagging behind, potentially waiting for the catch up. We think these numbers are a good example of the changing epidemiology of both diseases.

4. Lines 95-96. Given that effective treatment for both illness is the same, how important is differentiating anaplasmosis and ehrlichiosis cases clinically? Improving this rationale would help frame the analyses and usefulness of the results.

RESPONSE: We consider the recognition of disease-specific patterns (whether clinical or diagnostic) to be the core of clinical medicine and therefore relevant, regardless of whether the treatment is identical: anaplasmosis and ehrlichiosis differ in terms of their prognosis and complications, so that in the absence of diagnostic confirmation, the identification of potentially differentiating patterns would be desirable and in our view justified, especially even more so in areas which lack fast access to diagnostic tools.

5. Lines 110-111. Please provide more details on the systematic reviews as these form the basis for analyses here. For example, how were cases included in these reviews, what was the spatial and temporal scale of included case data, what information was included in each review?

RESPONSE: We added a line to the methods section to include additional criteria from systematic reviews for context “These reviews included all cases of anaplasmosis (published up to 27/Apr/2023)

and ehrlichiosis (published up to 26/June/2023) published in English, German, French, Italian or Spanish.”

6. Line 113. Coinfections are not often tested for in clinical settings and not necessarily reportable. However, coinfections potentially impact length and clinical course of the infections. How could the unintentional inclusion of coinfected cases have on the analysis?

RESPONSE: The original dataset of the two systematic reviews included co-infection. These cases were excluded for the analysis to reduce the risk for such contamination. This does not exclude the risk for the occasional co-infection entirely, or missed co-infections, as the data is observational and diagnostic methods have improved over the decades. However, many of the different co-infections would present in a different way, e.g. specifically target the CNS or present with different symptoms, such as rocky mountain spotted fever, or Lyme’s disease.

7. Line 123. How were parametric and non-parametric data identified?

RESPONSE: Continuous variables were assessed for normal distribution and were in all cases strongly skewed, with no normalcy, and no outliers that could not have been removed.

8. Lines 124-127. The results of this Kolmogorov-Smirnov test for differences in age structure does not appear to have been reported. Table 1 presents the results from a Pearson Chi-square test of differences in median age which is not the correct test for this comparison.

RESPONSE: We thank the reviewer for pointing this out. We added the information for age, as a Kruskal-Wallis rank sum test was used. We have also added the missing results for the Kolmogorov-Smirnov test to the results section.

9. Line 128. This statement appears to refer to data presented in Figure 2. However, case data stratified by age, gender, and immunosuppression status is not presented. Please reword this statement or update Figure 2 to provide concordance between methods and presented results.

RESPONSE: We corrected the typo and reworded the statement to correctly reflect both figures.

10. Lines 132-138. Please expand the Results beyond listing the tables and figure included. Highlight main takeaways presented. Some of this is currently in the Discussion.

RESPONSE: We expanded the results section with the most relevant takeaways.

11. Lines 132-133. A relatively large proportion of cases reported in the systematic reviews were not included in this analysis. Which inclusion criteria of the study removed most of these cases? Including a flow chart could illustrate the inclusion/exclusion process for the study.

RESPONSE: We have added a flow chart as supplementary Fig 1. The most common reason was the lack of individual data for the cases followed by restricting the cases to the USA.

12. Lines 166-170. It does not appear that comparing clinical and epidemiolocal data by immune status between anaplasmosis and ehrlichiosis cases was defined as a goal a prior to analyses. As such, the significance testing stratified by immune status (row 1 results presented in Table 1) would represent an ad hoc test, not the primary test. The significance of a Pearson’s chi-squared test on the 2x2 contingency table of immune status vs. case type should be presented. The results of these ad-hoc tests could still be presented, but with a correction for multiple comparisons.

RESPONSE: We thank the reviewer for pointing this out. Immune status was part of the original defined goal. We have added this to the methods sections for clarification.

13. Figure 1. I would suggest presenting the data as bar charts instead of the current density plots. Then it is easier for readers to compare case counts presented in Table 1 with those presented in the figure.

RESPONSE: We have changed the figure to include bar charts with HGA in the foreground for ease of comparison.

14. Tables 2-4. Given that the presence and number of asterisks are often used to denote statistical significance, I would suggest using different symbols to denote Fisher’s exact and Kruskal-Wallis tests in Tables.

RESPONSE: We have changed the signs used. It should now be more in line with convention.

16. Table 2. Please explain why the denominators vary across rows. For example, of the 153 immunocompetent cases of HGA, why is the denominator 124 for most symptoms? Was it because this data was not available?

RESPONSE: Indeed, the data was not available for these cases. It is unfortunately one of the features of case reports and case series that the data is not collected in a standardized form.

17. Please review wording throughout the manuscript as there are several grammatical errors. While they do not impair the understanding of the text, they reduce the appearance of high-quality work.

RESPONSE: It is not quite clear which sections should be adapted as reviewer 2 and the editor did not mention this at all. However, if there were specific sections, we would be happy to have a second look at them. 

Reviewer #2: This study addresses an important clinical topic; however, the review of the existing literature is insufficient. Several recent and relevant studies have not been cited, which limits the comprehensiveness and accuracy of the background presented. A more thorough literature review is necessary to appropriately contextualize the study findings and conclusions.

1) The term "Fingertips" in the title is misleading and should be modified accordingly. It is unclear what the authors intend to convey—perhaps they mean "characteristics"?

RESPONSE: Thank you for pointing this out, however, the title uses the forensic term “fingerprints” which would be a different way of using the term “characteristics”. We hope this clarifies why the term was used. We are open to change this if needed.

2) In the Introduction, line 66 is missing a reference. Additionally, the authors have not adequately reviewed the existing literature. A recent systematic review on this topic should be referenced and acknowledged: https://pubmed.ncbi.nlm.nih.gov/35889152/

RESPONSE: We are a bit surprised by this statement, as the introduction is based on the data from a systematic review by Schudl et al. (Fig. 2) and the statements made in line 66 are in line with that suggested review, as well as the presented systematic reviews. We have gladly added the suggested citation to line 66 as it always helps to have additional references for completion and the reader to look up.

3) Line 76: Both diseases are notifiable in all 50 states in the USA—not “in the most.”

RESPONSE: Thank you for pointing this out. We corrected this statement.

4) Lines 82–87: This section is not particularly illustrative. Moreover, epidemiological data are continuously changing. I suggest removing this part.

RESPONSE: We added this data to illustrate the changing epidemiology but also to illustrate how both diseases are increasing in number, which we think is helpful for the reader as our data is observational and retrospective. We would therefore rather keep this part for context.

5) Line 88: The above-mentioned systematic review also identified gastrointestinal symptoms as common in HGA, and this important detail should be added.

RESPONSE: We added the suggestion to mention “gastrointestinal symptoms”

6) Line 99: Please include a more recent reference such as: https://academic.oup.com/cid/advance-article/doi/10.1093/cid/ciaf171/8104456

RESPONSE: Thank you for the additional reference, we gladly added it.

7) Table 4 reports 29 cases of encephalitis, which appears to be an unusually high number. Please provide a reference for these cases. Encephalitis in HGA is considered extremely rare. Additional clarification or justification is needed—see the following reference: https://pubmed.ncbi.nlm.nih.gov/37489389/

RESPONSE: We do agree that this is an interesting result. That being said, these 29 cases are over an almost 30-year timespan, and to be fair the quality of the data is observational and retrospective, and although we excluded everyone with a secondary infection from the original data set, this does not exclude the occasional secondary infection being missed. Another point is, that hundreds of cases were excluded because they did not contain any detailed information, therefore it is unclear how many of these may have had a potential involvement of the CNS.

We did look at the provided reference https://pubmed.ncbi.nlm.nih.gov/37489389/, and found no data to support the statement provided in the that specific publication, as the referenced publication references https://pubmed.ncbi.nlm.nih.gov/17286796/ in regards to neurological symptoms. Though the conclusion from the provided reference by the reviewer is, that neurological involvement in HGA is still an area to be explored. There also appears to be a general problem with the references in the neurological section for HGA in https://pubmed.ncbi.nlm.nih.gov/37489389/, as reference 77 and 89 are both animal models investigating the role of cytokines and cellular immunity and not neurological data.

8) Discussion, lines 212–218: Please reference the recent systematic review on tick-borne infection-associated hemophagocytic lymphohistiocytosis (HLH), which provides critical insights into this life-threatening complication: https://pubmed.ncbi.nlm.nih.gov/38525759/. Anaplasma can cause secondary HLH, as described in this review, and the statement in this section should be revised to reflect this evidence

RESPONSE: Thank you for pointing this out, however we want to point out, that we did not claim that HGA does not cause HLH, but that HME causes sHLH significantly more often than HGA. Which is also reflected in the provided review. We have added the suggested reference to better reflect this and for additional resources to the reader.

---

## [Decision Letter · Decision Letter 1]

9 Jul 2025

Dear Dr. Kositz,

Thank you for submitting your manuscript to PLOS ONE. After careful consideration, we feel that it has merit but does not fully meet PLOS ONE’s publication criteria as it currently stands. Therefore, we invite you to submit a revised version of the manuscript that addresses the points raised during the review process.

We look forward to receiving your revised manuscript.

Kind regards,

Yong Qi

Academic Editor

PLOS ONE

Journal Requirements:

Reviewers' comments:

Reviewer's Responses to Questions

**Comments to the Author**

Reviewer #1: (No Response)

Reviewer #2: (No Response)

2. Is the manuscript technically sound, and do the data support the conclusions?

Reviewer #1: Yes

Reviewer #2: Yes

3. Has the statistical analysis been performed appropriately and rigorously?

Reviewer #1: Yes

Reviewer #2: I Don't Know

4. Have the authors made all data underlying the findings in their manuscript fully available?

Reviewer #1: Yes

Reviewer #2: Yes

5. Is the manuscript presented in an intelligible fashion and written in standard English?

Reviewer #1: Yes

Reviewer #2: Yes

Reviewer #1: Thank you to the authors for their revision of the manuscript and response to reviewer comments. The clarity of the manuscript has improved. I only have a few minor comments.

Line 76/Fig 1. There appears to be a discrepancy between the text (both diseases are notifiable in all 50 US states) and Figure 1 (several states are indicated as not notifiable). Please reconcile.

Figure 1. Given how similar the ranges in incidence are between the panels, I would suggest using the same cut points for color. The readers can compare relative magnitudes between panels.

Line 144. The ‘p’ is missing (i.e., should read p<0.001).

Line 150. There is a typo for “such”.

Table 2. I suggest the inclusion of a footnote explaining the reason for differences in denominators within columns (i.e., lack of standardized reporting). The authors explained this in their response to reviewers, but did not include this information in the revised document. I would also suggest the authors add this caveat explicitly to the limitations. Differential missingness of reported symptoms across diseases could bias the estimates proportions.

Reviewer #2: I would like to express my appreciation to the authors for their revisions. While not all of my concerns have been fully addressed, I wish to respect the authors’ autonomy, and I find the minor comments they chose not to incorporate to be acceptable.

However, one major concern remains regarding the discussion of Anaplasma encephalitis. The authors are quick to critique the case report and literature review conducted by researchers at the Mayo Clinic, yet they do not provide any substantiating evidence for the 29 cases cited in their results. Furthermore, the authors acknowledge the possibility of co-infection in these cases, which raises additional questions regarding the validity of the findings.

In my view, there are two appropriate courses of action: either (1) eliminate the discussion and corresponding results related to encephalitis entirely, or (2) retain the section but support it with a thorough and detailed review of the literature, specifically addressing all 29 cases included in the results.

**Do you want your identity to be public for this peer review?** For information about this choice, including consent withdrawal, please see our Privacy Policy

Reviewer #1: No

Reviewer #2: No

---

## [Author Response · Author response to Decision Letter 2]

23 Aug 2025

We cordially want to thank the two reviewers for taking their time to review our manuscript, their critical eyes, and their constructive remarks and input to the manuscript.

In the following, we will address the points listed by the two reviewers in chronological order and also address what we adapted in the manuscript following the comments from both reviewers.

Reviewer #1:

Thank you to the authors for their revision of the manuscript and response to reviewer comments. The clarity of the manuscript has improved. I only have a few minor comments.

Line 76/Fig 1. There appears to be a discrepancy between the text (both diseases are notifiable in all 50 US states) and Figure 1 (several states are indicated as not notifiable). Please reconcile.

RESPONSE: Thank you for pointing this out. We changed the “not notifiable” to “no data”, which is much more appropriate.

Figure 1. Given how similar the ranges in incidence are between the panels, I would suggest using the same cut points for color. The readers can compare relative magnitudes between panels.

RESPONSE: We do understand the request, however the reason for these different cutoffs, is that the map is a 1:1 recreation of the originals used by CDC including the cutoffs. The recreation was done because the original did not have the required dpi quality, and we would prefer to keep them as an identical copy of the original.

Line 144. The ‘p’ is missing (i.e., should read p<0.001).

RESPONSE: Corrected.

Line 150. There is a typo for “such”.

RESPONSE: Corrected.

Table 2. I suggest the inclusion of a footnote explaining the reason for differences in denominators within columns (i.e., lack of standardized reporting). The authors explained this in their response to reviewers, but did not include this information in the revised document. I would also suggest the authors add this caveat explicitly to the limitations. Differential missingness of reported symptoms across diseases could bias the estimates proportions.

RESPONSE: This is a good point, thank you very much. We added the line in the footnote. Additionally, we added the line “the differential missingness of reported symptoms across the two diseases could bias the estimated proportions” at line 254 and “or contradict” at line 257, to put more emphasis on the unfortunate lack of standardized reporting.

Reviewer #2:

I would like to express my appreciation to the authors for their revisions. While not all of my concerns have been fully addressed, I wish to respect the authors’ autonomy, and I find the minor comments they chose not to incorporate to be acceptable.

However, one major concern remains regarding the discussion of Anaplasma encephalitis. The authors are quick to critique the case report and literature review conducted by researchers at the Mayo Clinic, yet they do not provide any substantiating evidence for the 29 cases cited in their results. Furthermore, the authors acknowledge the possibility of co-infection in these cases, which raises additional questions regarding the validity of the findings.

In my view, there are two appropriate courses of action: either (1) eliminate the discussion and corresponding results related to encephalitis entirely, or (2) retain the section but support it with a thorough and detailed review of the literature, specifically addressing all 29 cases included in the results.

RESPONSE: We would like to point out, that we are not quick to criticize the literature review and not at all the case report itself, but the fact is that the reference used for .” Neurologic manifestations of anaplasmosis are less common than in other tick-borne diseases such as ehrlichiosis, Lyme disease, and Powassan virus infection [2,6]” and “Neurologic symptoms and complications in HGA are rare [2]” where specifically ref 2 is “Ismail, N.; McBride, J.W. Tick-Borne Emerging Infections.” and the paragraph on“Human Granulocytic Anaplasmosis Neurologic Features” in Ismael et al on neurological symptoms cites:

- Choi KS, Scorpio DG, Barat NC. Msp2 variation in Anaplasma phagocytophilum in vivo does not stimulate T cell immune responses or interferon-gamma production. FEMS Immunol Med Microbiol 2007;49:374–86.

- Horowitz HW, Marks SJ, Weintraub M, et al. Brachial plexopathy associated with human granulocytic ehrlichiosis. Neurology 1996;46:1026–9.

- Akkoyunlu M, Fikrig E. Gamma interferon dominates the murine cytokine response to the agent of human granulocytic ehrlichiosis and helps to control the degree of early rickettsemia. Infect Immun 2000;68:1827–33.

This might have been citation oversight by Ismael et al. Nevertheless, none of these three citations provide any prevalence or incidence data on the involvement of the CNS. Therefore, this reference does not hold up.

We do acknowledge that ref 6 “Bakken, J.S.; Dumler, J.S. Human Granulocytic Anaplasmosis” references the respective data, with 5 cases over 6 years from Dahlgren et al. However, they had access to the national database of the CDC, whereas published case reports can suffer from a “this is interesting, let’s publish it” bias as seen by the much lower numbers of published cases vs. the national database. Additionally, the amount of our encephalitis cases is over 30-year period which roughly corresponds to the amount found by Dahlgren et al. over their 6 years, i.e. a case per year. Furthermore, we do not claim that encephalitis is common in HGA, but that it may be more frequent in HGA when compared to HME, although still an overall small number.

We reviewed the cases, excluded those with other potential causes which reduced the immunocompetent cases to 22, which would be a p of 0.04, primarily based on confusion, as most additional data was lacking (liquor, EEGs, etc.) for a more solid definition of encephalitis (based on the 2013 criteria).

Because of the long-time frame and potentially different definitions of encephalitis and the often missing additional information in the case reports, we decided to remove the encephalitis line from the results, the table and the discussion.

We added this line to the discussion “Unfortunately, the data from these case reports were insufficient to provide a reliable estimate on the difference of cases with possible encephalitis between the two diseases, and therefore this point remains unresolved.” for clarification.

---

## [Decision Letter · Decision Letter 2]

6 Oct 2025

Comparison of the epidemiological and clinical fingerprints of Human Granulocytotropic Anaplasmosis and Human Monocytotropic Ehrlichiosis in the United States

PONE-D-25-14987R2

Dear Dr. Kositz,

We’re pleased to inform you that your manuscript has been judged scientifically suitable for publication and will be formally accepted for publication once it meets all outstanding technical requirements.

Kind regards,

Yong Qi

Academic Editor

PLOS ONE

Additional Editor Comments (optional):

Reviewers' comments:

Reviewer's Responses to Questions

**Comments to the Author**

Reviewer #1: All comments have been addressed

Reviewer #2: All comments have been addressed

2. Is the manuscript technically sound, and do the data support the conclusions?

Reviewer #1: Yes

Reviewer #2: Yes

3. Has the statistical analysis been performed appropriately and rigorously?

Reviewer #1: Yes

Reviewer #2: Yes

4. Have the authors made all data underlying the findings in their manuscript fully available?

Reviewer #1: Yes

Reviewer #2: Yes

5. Is the manuscript presented in an intelligible fashion and written in standard English?

Reviewer #1: Yes

Reviewer #2: Yes

Reviewer #1: (No Response)

Reviewer #2: The paper is acceptable in its current form for acceptance. I do not have any further comments. Congratulations and thank you for reporting this interesting data

**Do you want your identity to be public for this peer review?** For information about this choice, including consent withdrawal, please see our Privacy Policy

Reviewer #1: No

Reviewer #2: No

---

## [Editor Report · Acceptance letter]

PONE-D-25-14987R2

PLOS ONE

Dear Dr. Kositz,

I'm pleased to inform you that your manuscript has been deemed suitable for publication in PLOS ONE. Congratulations! Your manuscript is now being handed over to our production team.

Kind regards,

on behalf of

Dr. Yong Qi

Academic Editor

PLOS ONE